# Biological Activities and Polyphenol Content of Qi Cha Tea^®^, a Functional Beverage of White Tea Containing Botanicals and Dry Botanical Extracts with European Health Claims

**DOI:** 10.3390/plants12183231

**Published:** 2023-09-11

**Authors:** Jean Michel Maixent, Meriam Belaiba, Olivier Pons, Enora Roulleau, Jalloul Bouajila, Jean-Marc Zeil

**Affiliations:** 1Pierre Deniker Clinical Research Unit, Henri Laborit University Hospital, University of Poitiers, F-86000 Poitiers, France; tessa.roulleau@gmail.com; 2Laboratory Impact of Physical Activity, Health (I.A.P.S.) Toulon University, F-83000 Toulon, France; olivier.pons34310@gmail.com; 3School of Sciences, Poitiers University, F-86000 Poitiers, France; 4Laboratoire de Génie Chimique, University Paul Sabatier, CNRS, INPT, UPS, F-31100 Toulouse, France; belaibameriam@gmail.com (M.B.); jalloul.bouajila@univ-tlse3.fr (J.B.); 5Thés de la Pagode, 4, Avenue Bertie Albrecht, 75008 Paris, France; jm.zeil@thesdelapagode.fr

**Keywords:** *Camellia sinensis*, acacia gum, infusion beverages, organic, health claims

## Abstract

Infusions of *Camellia sinensis* leaves have been known for their health benefits. The Bio Concentrate Assets^®^ (ABC) method is a method of enriching organic infusion leaves (from *Camellia sinensis*) with organic dry and concentrated extracts using organic acacia gum, and its application to white tea has provided Qi cha tea^®^. In the present study, the content of tea polyphenols and caffeine, and the biochemical properties of Qi cha tea^®^ and its botanical constituents (elderberry, tulsi, *Echinacea purpurea*, orange peel, lemongrass, and acacia gum) were assessed. Antioxidant and cell viability activities were determined by the 1,1-diphenyl-2-picrylhydrazyl (DPPH) assay and MTT (3-(4, 5-dimethyl thiazol-2-yl)-2, 5-diphenyl tetrazolium bromide) assay in human Caco-2 and HCT-116 cell lines, and ascorbic acid and tamoxifen, respectively. The caffeine and polyphenol composition of Qi cha tea^®^ was modified with less caffeine and gallic acid and more epigallocatechin gallate (EGCG) than the original white tea. The majority of the tested botanical samples including Qi cha tea^®^ at 50 µg/mL show similar antioxidant activities, with the exception of *Echinacea purpurea*. The greatest effect was found for white tea. The antioxidant power of the Qi cha tea^®^ (90% at 50 µg/mL for pressurized liquid extraction (PLE) was divided by approximately a factor of two (61% at 50 µg/mL for pressurized liquid extraction products (PLEP)), which corresponds to the 48.3% (mass/mass) white tea original content in the Qi cha tea^®^. Qi cha tea^®^ shows the lowest cytotoxic activity in the viability of the two cell lines when compared to white tea. The application of the ABC method to Qi cha tea^®^ using various botanicals and dry extract with acacia gum as blinder has allowed the development of a new innovative functional health beverage that complies with European health claims.

## 1. Introduction

Plant-based foods such as teas, coffee, spices, fruits, and others have been extensively studied for their potential health benefits [1]. Leaves from *Camellia sinensis* (*C. sinensis*) are used to make the second most widely consumed beverage in the world after water, and it is a good example of a beverage associated with health [2]. In China, water infusions of *C. sinensis* leaves have been known for their traditional benefits [3]. Daily tea consumption by generally healthy adults has been associated, in a dose-dependent manner, with lower risk of mortality [4]. The bioactive components of the leaves and the teas brewed from them are the subject of intensive research for phytochemical content characterization [5,6]. The molecular mechanisms of tea’s health function in the context of chronic diseases have been reported to include their alkaloids and catechins, epicatechin, and their metabolites [4,5,6,7] (Figure 1). For some diseases, a therapeutic approach has been achieved with traditional medicinal plants with the patented *Ginkgo biloba* extract (EGb 761) [8]. However, the use of plant-based foods for prevention of these human chronic diseases in the healthy population is not always possible, as the evidence provided is insufficient to establish a scientifically based substantiation needed for a therapeutic claim related to a food and/or its constituents [2,9]. 

In 2006, the European Union (EU) adopted a regulation on the use of nutrition and health claims for foods (the Nutrition and Health Claims Regulation (EC) No 1924/2006) [10,11,12]. However, EU health claims (for health in relation to a function of the body (Article 13 claims)) for botanicals remain as “on hold” health claims [11,12]. An innovative method afforded by the Bio Concentrate Assets^®^ (ABC) patent have been developed. The ABC process is an exclusive method of enriching organic infusion leaves with organic dry and concentrated extracts using organic acacia gum and its application of this method has provided us with nine new beverages, the first one being Qi cha tea^®^. Qi cha tea^®^ is a mixture of white tea and elderberry, tulsi, *Echinacea purpurea*, orange peel, lemongrass and acacia gum. Since some steps of the ABC process could alter the biological activities of the new tea beverage, as there is adsorption of the active ingredients with the use of the hydroalcoholic solution with acacia gum and then evaporation of ethanol solution by drying. No information on the use of these processes with teas has been reported, to the best of our knowledge. This article aims to evaluate qualitatively and quantitatively the major constituents of the first beverage produced with the ABC process, i.e., Qi cha tea^®^, by comparing with commercially available chemicals. Antioxidant activity of *C. sinensis* is related to its phenolic compounds. The preservation of the water-soluble phenolic compounds are essential for *C. sinensis* and for infusions issued from the ABC process. Therefore, the content of gallic acid, tea catechins, and caffeine was evaluated using high-performance liquid chromatography with diode-array detection (HPLC–DAD). These constituents were evaluated in two extracts obtained by water infusion (WI) and pressurized liquid extraction (PLE). As well, “in vitro” antioxidant and cytotoxic activities in two cell lines were determined by 2,2-Diphenyl-1-picrylhydrazyl (DPPH) assay and 3-(4,5-Dimethyl-2-thiazolyl)-2, 5-diphenyl-2H-tetrazolium bromide (MTT) assay, respectively. These two human targets were chosen to evaluate the antioxidant potential of the extracts, which expresses the beneficial effect on health, and the anti-cancer potential against two lines of colon cancer, which is directly related to food consumption.

## 2. Materials and Methods

### 2.1. Chemicals, Water, and Devices

All chemicals used were purchased from Sigma-Aldrich-Fluka (Saint-Quentin, France) and stored under optimum conditions as indicated by the manufacturer. 

For the device, the HPLC analysis was carried out using an ultimate 3000 pump-Dionex and Thermos Separation product detector DAD model (Thermo Fisher Scientific, Waltham, MA, USA). Separation was performed with an RPC18 reversed-phase column (Phenomenex, Le Pecq, France). The samples were filtered through a Millex syringe filter (Sigma Aldrich, Saint-Quentin-Fallavier, France).

Drinking water was used for the ABC process performed in France, where drinking water is defined as water that can be consumed without risk to health. This definition of drinking water is governed by strict standards that set limits for harmful substances that drinking water must not exceed. The quality criteria for water intended for human consumption are laid down in the French Public Health Code (CSP), which covers “all water used in food businesses for the manufacture, processing, preservation or marketing of products”.

Organic white tea, dried parts (elderflower, orange peel, tulsi, *Echinacea purpurea* dry extract, and lemongrass), and natural aromas of bergamot and orange were obtained from Thé de la Pagode, Paris, France.

### 2.2. Preparation of Qi Cha Tea^®^ by the Bio Concentrate Assets^®^ (ABC) Method

A predetermined quantity of plant fragments or whole plants was introduced in a 2 L balloon as a carrier. These plants were then homogenized by rotation (using a rotavapor at room temperature). The desired quantity of dry extracts was then introduced and these solids were mixed by rotating the flask for at least 10 min. At the same time, a 3% acacia gum hydroalcoholic solution was prepared: water, 100% ethanol, and acacia gum were added to a beaker fitted with a magnetic rod. The solution was mixed for about 1 h until the acacia gum was perfectly dissolved. Part of the hydro-alcoholic solution thus obtained was removed and the various liquid flavors were introduced into the remaining solution. The solution was then thoroughly mixed. Part of the 3% acacia gum hydroalcoholic solution, containing the flavorings, was then introduced drop by drop (or sprayed) onto the initial carrier preparation contained in the 2 L flask. The solution was stirred by rotation as it was wetted. The mixing operations were repeated until all of the 3% acacia gum hydroalcoholic solution was introduced. The dry extract powder then adhered to the plant fragment support by the sticky effect of acacia gum. Part of the water and all the ethanol were then removed by drying to regain a standard moisture content for this type of infusion. Ethanol was also used to accelerate the drying process as a surface tension gradient. Drying was carried out by blowing dry air over the preparation in the oven at 40 °C, during which time it was continuously stirred. 

Application of the ABC method to white tea with 5 organic phytochemicals and natural aromas produces a new infusion plant called Qi cha tea^®^.

The composition of Qi cha tea^®^ is shown in Table 1. 

### 2.3. Preparation of Water Infusion Products (WIP) and Pressurized Liquid Extraction Products (PLEP)

A total of 100 mL of boiled water was added to 10 g of each sample (Qi cha tea^®^ or botanicals), and infused for 30 min. The mixture was filtered through a filter paper. The resulting filtrate was evaporated using a rotary evaporator to provide WIP. The solid mass was weighed to estimate the extraction yield. 

PLE was performed by the Dionex extraction system (ASE 100, Sunnyvale, CA, USA). The extraction cell was filled with 10 g of the sample using water. The extraction conditions were 80 °C and 110 bar. The solvent was evaporated using a rotary evaporator under a vacuum at 35 °C (IKA, Staufen, Germany).

### 2.4. Chromatographic Fingerprint Analyses by High-Performance Liquid Chromatography Coupled with Diode-Array Detector (HPLC–DAD) Analysis

Analytical HPLC gradient elution methods were used for Qi cha tea^®^ or botanicals [13]. Standard solution named G-016 (Sigma Aldrich, Saint-Quentin-Fallavier, France) contained: caffeine, EGCG, epicatechin (EC), epigallocatechin (ECG), gallocatechin (GC), gallocatechin-3-gallate (GCG), catechin-3-gallate (CG), epigallocatechin (EGC), and gallic acid (GA). The dried extracts were injected at a concentration of 20 µg/mL of acetonitrile and water (20:80 *v*/*v*). The contents in the samples were determined after chromatographic fingerprint analyses by high-performance liquid chromatography coupled with diode-array detector (HPLC–DAD) from Thermo Fisher Scientific, USA) as previously reported by Dawra et al. [13]. The HPLC analysis was carried out using an ultimate 3000 pump-Dionex and Thermos Separation product detector DAD model. The separation was carried out on an RPC18 reversed-phase column (Phenomenex, Le Pecq, France). The samples were filtered through a Millex syringe filter (Sigma Aldrich, Saint-Quentin-Fallavier, France). The identification of the compounds by HPLC–DAD was based on comparison of the retention times and the DAD spectra at the maximum absorbance of each compound to those obtained with the co-injection of commercial polyphenols standards. 

Standard solutions were separately prepared at concentrations of 50, 20, 10, 5, and 1 µg/mL using (20:80 *v*/*v*) acetonitrile/water.

### 2.5. Free Radical Scavenging Activity: DPPH Test

Antioxidant scavenging activity was studied using the 1,1-diphenyl-2-picrylhydrazyl free radical (DPPH) with some modifications [14]. About 20 µL of various dilutions (water) of each sample was mixed with a 0.2 mM methanolic DPPH solution. After 30 min of incubation at 25 °C, the absorbance at 524 nm was recorded as A (sample). For A (blank), the same experimentation was applied for a solution devoid of the test material and then the absorbance was recorded. Then, for each solution the free radical, scavenging activity was calculated as percent inhibition as the following equation:% Inhibition = 100 × [(A (blank) − A (sample))/A (blank)]

The IC_50_ is the concentration required for the test material to cause a 50% decrease in DPPH concentration. Ascorbic acid at 5.9 μg/mL was used as reference. All the measurements were performed in triplicate.

### 2.6. Cell Viability Evaluation

The cell viability was evaluated by MTT (3-(4, 5-dimethyl thiazol-2-yl)-2, 5-diphenyl tetrazolium bromide) assay. Briefly, MTT is a yellow tetrazolium salt. In the biological system, the succinate dehydrogenase, a mitochondrial enzyme of the viable cells, reduces the tetrazolium to dark blue formazan precipitate as previously described [13,14]. The quantity of this product is proportional to the quantity of active cells. The cell viability (cytotoxic activity) of each compound was determined in the human Caco-2 and HCT-116 cell lines (American Type Culture Collection, Manassas, VA, USA). In a 96-well microplate, cells were distributed at 13 × 10^3^ cells/well for HCT116 and 12 × 10^3^ cells/well for Caco-2 in 100 µL of a suitable culture medium. After 24 h of incubation at 37 °C, 100 µL of each extract diluted in the medium after being solubilized in DMSO was added to 100 µL of the corresponding culture medium; RPMI (RPMI 1640, Thermo Fisher Scientific, Illkirch, France) for HCT-116, or DMEM (Advanced DMEM, Thermo Fisher Scientific). The final concentration of the extract in each well was 50 μg/mL. The plate was then incubated for 48 h at 37 °C and the cytotoxic potential of the tested samples was evaluated by the MTT assay. After removing the supernatant, cells were treated with 50 µL of MTT solution, and the plate was incubated for 40 min at 37 °C; then, MTT was eliminated and 80 µL of DMSO was added. The absorbance was measured at 605 nm using a microplate reader (Multiskan Go, F1-01620, Thermo Fisher Scientific, Vantaa, Finland). The inhibition percentage of cell proliferation was calculated as follows: % inhibition = 100 × (Ablank-Asample/Ablank). The mitochondrial reduction of MTT to formazan was used to determine the cytotoxic effect of those compounds. Tested samples were suspended in the dimethyl sulfoxide (DMSO) and then diluted, so the concentration of DMSO did not exceed 1% in the mixture. Tamoxifen was used as a positive control. The test was performed in triplicate.

### 2.7. Statistical Analysis

All data were expressed as mean ± standard deviation of duplicate measurements. 

One-way analysis of variance (ANOVA) was used for the significance calculation using the Statistical Package for the Social Sciences (SPSS) 20.1 (Version IBM. 20.0. 2004, San Francisco, CA, USA). 

Statistical differences between the solvents used in the study were estimated using Tukey’s test. The confidence limits were set at *p* ≤ 0.05.

## 3. Results

All the raw materials as well as the final tea were used to prepare extracts. These latter extracts were evaporated to dryness to estimate the yields, perform the chromatographic analyzes by HPLC–DAD, and evaluate the antioxidant and cytotoxic activities.

### 3.1. Efficacy for Preparing WI Products (WIP) and PLE Products (PLEP)

The typical chromatograms of Qi cha tea^®^ water extract obtained by both methods (water infusion WI or PLE water extraction) are shown in Figure 2. Extract yields from Qi cha tea^®^ and botanicals to prepare WIP and PLEP are listed in Table 1. 

As compared to WIP, PLEP shows the improved extraction yields in the case of elderberry, tulsi, white tea, and lemongrass samples. For *Echinacea purpurea*, the extraction yield was about 100% because the sample was already a dried extract. The yields of WIP from Qi cha Tea^®^ and orange peel are higher than those of PLEP. This result may be due to the presence of terpenes from essential oils, for example, which are recovered in the infusion at the surface of the water phase (essential oil extraction principle), whereas the PLEP preparation method does not allow extract of them under the effect of pressure given the large difference in polarity.

### 3.2. HPLC–DAD Analysis

Analysis of Qi cha tea^®^ and botanical samples using HPLC–DAD was conducted. All tests were performed in duplicate. The results for Qi cha tea^®^ and its constituents are shown in Table 2. The quantification of the following components (G1–G8): [caffeine (G1)—(−)–epigallocatechin-3-gallate (EGCG, G2)—(+)-catechin (G3)—(−)-epicatechin (G4)—(−)-epicatechin-3-gallate (G5)—(−)-gallocatechin (G6)—(−)–gallocatechin-3-gallate (G7)—(−)–catechin-3-gallate (G8)], gallic acid (GA), and epigallocatechin (EGC) levels of the Qi cha tea and botanicals were compared. A significant variation in the HPLC–DAD profiles of WIP and PLEP Qi cha tea^®^ was observed (Figure 2). In the analyzed samples, it is observed that the extracts obtained by PLE show an improvement of 113, 13, 1.7, 15, 4.6, and 3.6 times of GA, EGC, G1, G2, G4, and G6 for Qi cha tea^®^. For other samples, the highest content of compounds from the catechin family are observed in samples of Qi cha tea^®^ and also in organic white tea, with amounts of 5173 mg/kg for EGC and 9095 mg/kg for G1 in Qi cha tea^®^, and 4089 mg/kg for compound G3 in organic white tea. *Echinacea purpurea* also contained an important quantity of the component G4 (9653 mg/kg). The most important difference is observed in Qi cha tea^®^ extract PLEP, when seven molecules are quantified, and this is strong evidence that PLE enhances the extraction and isolation of molecules compared to the classic direct infusion method. The Qi cha tea^®^ analysis obtained with further PLE reveals high levels of GA (1470 mg/kg), EGC (5173 mg/kg), and G1 (9095 mg/kg). 

The comparison of Qi cha tea^®^ WIP to the original WIP white tea polyphenols and caffeine shows that the composition of caffeine, catechin, and epicatechin is modified, resulting in less caffeine and gallic acid, and the presence of EGCG (G2). The EGCG (G2) was not detected in the white tea that was used.

### 3.3. Antioxidant Capacity and Cytotoxic Activity of Qi Cha Tea^®^ and Its Components

Antioxidant properties as determined by the method using DPPH and cytotoxic activity of Qi cha tea^®^ and botanicals in human Caco-2 and HCT-116 cell lines are shown in Table 3.

#### 3.3.1. Antioxidant Activity

Qi cha tea^®^ inhibition strength is increased in PLEP and doubles in percentage from 32 to 61% as compared to Qi cha tea^®^ WIP. This result indicates that the method to prepare PLEP enhances antioxidant activity and can ameliorate bioactive molecule isolation. The majority of Qi cha tea^®^ and botanicals tested at the concentration 50 µg/mL show measurable antioxidant activity, with the exception of *Echinacea purpurea*. The greatest activity is found in white tea PLEP (90% of inhibition at 50 µg/mL and an IC_50_ equal to 29 µg/mL). Tulsi PLEP was also considered to be a very potent antioxidant (82% of inhibition at 50 µg/mL). 

The antioxidant power of the Qi cha tea^®^ (90% at 50 µg/mL for PLEP) was divided by approximately a factor of two (61% at 50 µg/mL for PLEP), which corresponds to the 48.3% (mass/mass) white tea content in the Qi cha tea^®^.

Orange peel and *Echinacea purpurea* antioxidant activities are low: 7, 14, 6, and 21% for WI and PLE, respectively. The Qi cha tea^®^ presents an important antioxidant level towards DPPH. This is also observed for tulsi and white tea. Lemongrass has a medium level of antioxidant action: 40% at 50 µg/mL.

#### 3.3.2. Cytotoxic Activity

The results for cytotoxic activity are shown in Table 3 for PLEP. Qi cha tea^®^ shows the lowest cytotoxic activity in both cell lines used when compared to white tea. The results demonstrate that the percentage of viability is high and varies from 63 to 80% at 50 µg/mL in the Caco-2 cell line for the first six samples tested (Table 3). The viability of the lemongrass PLEP in Caco-2 cells is about 49%. In HCT-116 colon cancer cells, cytotoxicity (100% viability) of tested PLEP samples, including Qi cha tea^®^, vary from 12 to 28%. 

## 4. Discussion

Many processes such as encapsulation nanoparticles have been published for tea catechin preservation [15,16,17]. A white tea formulation called Qi cha tea^®^ was created using the new process termed as the ABC process (Table 4) [18]. 

The technological advantages of the ABC method include the ability to adsorb the active ingredients more uniformly due to the use of the hydroalcoholic solution with added acacia gum, which could modify the new beverages in a way to reduce their biological activities and chemical composition. Therefore in this study, we analyzed the chemical composition, the antioxidant activity, and the cell viability or cytotoxicity activities of Qi cha tea^®^ and all botanical constituents of Qi cha tea^®^ (a mixture of white tea and elderberry, tulsi, *Echinacea purpurea*, orange peel, lemongrass, and acacia gum). Two methods of analysis were used. The data indicate that the PLE from Qi cha tea^®^ has higher levels of tea polyphenols and caffeine than Qi cha tea^®^ WI (Table 2). Other compounds were present only in either WIP or PLEP. This difference can be explained by a competition between all the substances present in the plant matrix and their affinities with respect to the solvent on the one hand and the influence of the extraction parameters (agitation, temperature and extraction time for infusion pressure, and temperature). Indeed, in addition to these tracked substances, there are other metabolites that are extracted. This competition generates variable extract yields and a risk of hydrolysis, oxidation, or reduction under the effect of temperature.

Caffeine is a major compound in tea [19]. Caffeine content from white tea WIP was found to be one of the most abundant compounds in our study according to a previous study of Pan et al. [20]. The catechins EGCG, EC, ECG, and EGC were the major polyphenol compounds, according to Pan et al. [20]. EGCG was not detected in the white tea WIP but detected in the Qi cha tea^®^ WIP. Using 6.5% orange peel for the process of Qi cha tea^®^ and only being detected in PLEP could not explain the presence of EGCG in the Qi cha tea^®^ WIP. Our results with undetectable ECGC in the white tea WIP are similar to results of AlHafez et al. [21]. Thus, as the water infusion composition is close to that brewing tea beverages, the ABC process plays a role in polyphenol and caffeine concentrations of Qi cha tea^®^ by reducing caffeine content and increasing EGCG content. 

EGC is present in Qi cha tea^®^ (WIP or PLEP) and comes mainly from white tea (1054–1370 mg/kg) or tulsi (414–1027 mg/kg), and, to a lesser extent, from lemongrass (255 mg/kg). Its PLE extraction is the highest (5173 mg/kg). This high recovery of PLEP can be explained by the influence of parameters (higher temperature and pressure) in the PLE.

Determination of the biochemical activity of Qi cha tea^®^ confirms an important antioxidant level in the DPPH assay. This was also observed for tulsi and white tea. High antioxidant power is likely related to high level of molecules present in each sample. There are several principles of methods for antioxidants methods. DPPH radical scavenging is an easy spectrophotometric method and a useful method with regard to measuring the radical scavenging capacity of food constituents and plant extracts [22]. Furthermore using three antioxidant methods, scavenging DPPH ranked with the highest antioxidant activity among these three methods, according to total phenolic contents [23]. More specifically, DPPH radical scavenging activity was used to characterize different teas from green, white, oolong, and black teas [24]. 

Evaluation of cytotoxic effects on HCT-116 and Caco-2 cells was comparable between the Qi cha tea^®^ and the original sample extracts. The evaluation of biological activity of Qi cha tea^®^ and its components are in good agreement with previous studies [25,26]. White tea and *Echinacea purpurea* show comparable cytotoxicity activity (28%). Hajiaghaalipour et al. [27] cited that an extract of white tea inhibited the proliferation of HT-29 cells with an IC_50_ of 87 μg/mL. In another work, the IC_50_ of white tea extract is 324 µg/mL against the MFC-7 cell line [28]. These results prove that our extracts are much more active against the two cell lines studied.

A previous study evaluated the anti-proliferative property of the white tea and demonstrated that white tea containing an important quantity of EGC seems to have the better anti-cancer effect than other teas [29]. It has been shown that one of the anti-cancer processes of growth inhibitory activity of tea may involve the catalytic regulation of the P450 enzymes and glucuronosyltransferase [30].

In the literature, catechins (flavan-3-ols), which are considered low-cost applicable phytochemicals, are the major green tea polyphenols [24]. EC, EGC, ECG, and EGCG are major components in little fermented teas such as white tea. EGCG from green teas is the most abundant and biologically active catechin [31,32]. Based on decades of research, EGCG has received considerable attention for its inhibitory activities against cancer initiation, promotion, and progression [26]. Other catechins, such as ECG and EGC, have been shown to have similar, albeit lower, activities in numerous studies [3,4]. Concerning white teas, the levels were not considered to be very high in a previous study [20]. In another study conducted on aqueous methanol (70%) extracts, DPPH radical scavenging activity and antiproliferative effects of white tea against colorectal cancer cell line HCT-116 were most likely due to its catechin content [31,33]. The determination of antioxidant and others biochemical properties of these food formulations may lead to explanations for others health claims [2,11,34]. Table 5 presents the formulation of Qi cha tea^®^, and the function health claim wording for respiratory and immune systems (ID2466 and ID2366, respectively) with tulsi, elderberry, and *Echinacea purpurea* respectively (18). In this study, the formulation of Qi cha tea^®^, (a mixture of white tea and elderberry, tulsi, *Echinacea purpurea*, orange peel, lemongrass, and acacia gum) is presented. Its biological activities were characterized “in vitro” for its antioxidant activity and for its cytotoxicity with human cells and the associated health claims that result from an original and organic tea beverage processed according to the patented ABC method innovation. According to the “on hold” European botanical heath claims, new health claims could be associated with botanical health claims. It will be of interest to explore its application to various botanicals from the European botanical list to improve human healthcare with the ABC method. 

## 5. Conclusions

The Bio Concentrate Assets^®^ (ABC) method has many advantages and offers a significant breakthrough in the production of teas and botanical infusions for health and wellness. Qi cha tea^®^ is the white-tea-based formulation produced by this innovative method. Qi cha tea^®^ maintains its antioxidant activity and shows the lowest cytotoxic activity in the viability of the two cell lines when compared to white tea. In conclusion, Qi cha tea^®^ (a mixture of white tea and elderberry, tulsi, *Echinacea purpurea*, orange peel, lemongrass, and acacia gum) has allowed the development a new innovative functional health beverage that complies with European health claims.

## 6. Patents

European patent EP 3501291, PROCÉDÉ DE FABRICATION D’INFUSIONS OU DE THÉS CONTENANT DES ACTIFS D’ORIGINE VÉGÉTALE, ANIMALE OU MINERALE. Owner of the patent: Thés de la Pagode, 4 Avenue Bertie Albrecht, 75008, Paris, France.

## Figures and Tables

**Figure 1 plants-12-03231-f001:**
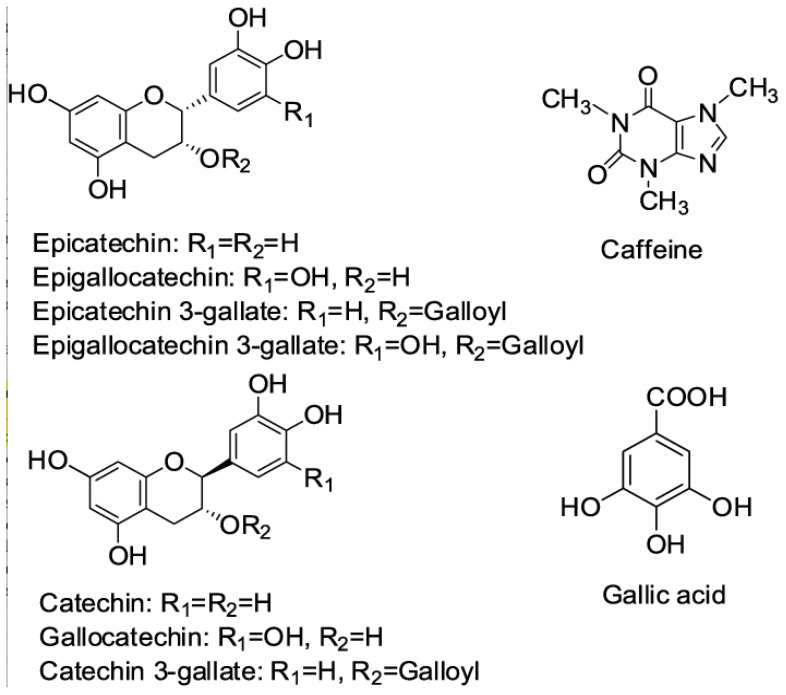
Chemical structures of major phytochemicals in green and white teas.

**Figure 2 plants-12-03231-f002:**
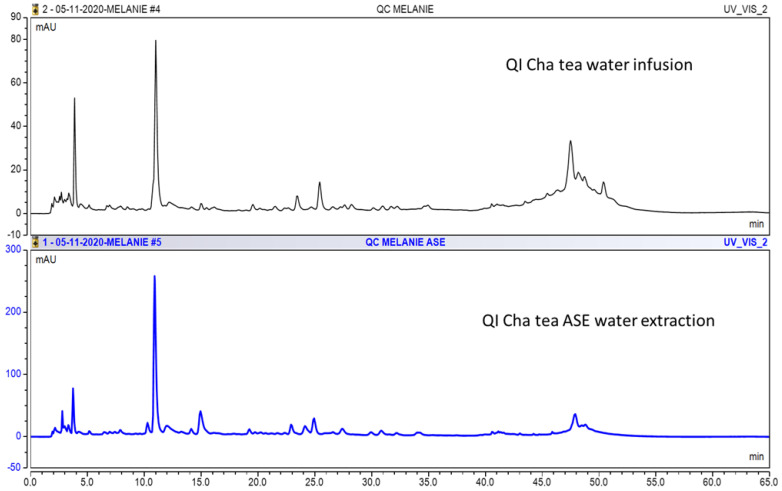
Typical HPLC chromatograms (280 nm) of WI products (WIP) and PLEP products (PLEP) from QI cha tea^®^ water infusion (upper) and QI cha tea^®^ PLE (ASE) water extraction (lower).

**Table 1 plants-12-03231-t001:** Extract yields for Qi cha Tea^®^ and botanicals in preparation of water infusion (WI) and pressurized liquid extraction (PLE).

	Yields (%)	
Sample	WIP	PLEP
Qi cha Tea^®^	21	16
Elderberry	11	27
Orange peel	29	24
Tulsi	16	23
White tea	15	25
*Echinacea purpurea*	101	98
Lemongrass	12	22

**Table 2 plants-12-03231-t002:** Content of tea polyphenols and caffeine in water infusions products (WIP) and pressurized liquid extraction products (PLEP).

		Content * (mg/kg)									
Sample		GA	EGC	G1	G2	G3	G4	G5	G6	G7	G8
Y = ax + b *	a	0.2948	0.0614	0.1104	0.3085	0.0442	0.0966	0.0966	0.1659	0.1291	0.1727
	b	1.6153	−1.9181	−1.5389	−4.1037	−0.6985	−0.5171	−0.5171	−2.7357	−1.4601	2.5116
R2		0.9992	0.9987	0.9995	0.9946	0.9975	0.9991	0.9958	0.9955	0.9933	0.9973
Qi cha tea^®^ WIP		13	384	585	295	331	338	-	238	-	-
Qi cha tea^®^ PLEP		1470	5173	9095	1384	-	1813	1477	862	-	-
Elderberry WIP		-	-	384	-	-	-	-	-	-	-
Elderberry PLEP		-	-	-	-	1137	-	-	-	-	-
Orange peel WIP		-	-	-	-	-	-	1066	-	-	-
Orange peel PLEP		-	-	-	276	-	1454	1249	-	299	299
Tulsi WIP		-	1027	115	-	-	835	157	-	148	148
Tulsi PLEP		-	414	-	-	-	-	-	-	-	-
White tea WI P		1706	1370	3701	-	4089	1057	481	-	-	-
White tea PLEP		-	1054	439	-	-	-	-	279	-	-
*Echinacea purpurea* WIP		-	-	1012	-	-	-	-	-	-	-
*Echinacea purpurea* PLEP		-	-	-	-	-	9653	-	4339	-	-
Lemongrass WIP		-	255	-	-	-	-	77	140	-	-
Lemongrass PLEP		225	-	-	-	-	-	-	-	-	-

* A linear equation of the form (Y = ax + b) was used to define the slope a and b on the Y-axis. R2 > 0.9 mean large positive linear association. Gallic acid (GA), epigallocatechin (EGC), caffeine (G1)—(−)-epigallocatechin 3-gallate (G2)—(+)-catechin (G3)—(−)-epicatechin (G4)—(−)-epicatechin-3-gallate (G5)—(−)-gallocatechin (G6)—(−)-gallocatechin 3-gallate (G7), and (−)-catechin 3-gallate (G8) levels in Qicha tea^®^, tea, and botanicals using HPLC–DAD. “-”: not determined.

**Table 3 plants-12-03231-t003:** Antioxidant and cytotoxic activities of Qi cha Tea^®^ and botanicals (50 µg/mL).

Sample	Antioxidant Activity	Cytotoxic Activity *	
	(DPPH Assay)	(MTT Assay)	
	DPPH	Caco-2	HC-T116
	Inhibition%	Viability%	Viability%
	(IC_50_ µg/mL)		
Qi cha tea^®^ WIP	32 ± 2		
Qi cha tea^®^ PLEP	61 ± 2 (39)	77 ± 6	85 ± 3
Elderberry WIP	43 ± 3		
Elderberry PLEP	58 ± 2 (45)	64 ± 5	86 ± 1
Orange peel WIP	7 ± 2		
Orange peel PLEP	14 ± 2	80 ± 8	82 ± 2
Tulsi WIP	52 ± 2 (48)		
Tulsi PLEP	82 ± 4 (31)	72 ± 2	83 ± 2
White tea WIP	59 ± 3 (41)		
White tea PLEP	90 ± 2 (29)	63 ± 11	72 ± 1
*Echinacea purpurea* WIP	6 ± 6		
*Echinacea purpurea* PLEP	21 ± 3	66 ± 2	72 ± 1
Lemongrass WIP	26 ± 1		
Lemongrass PLEP	40 ± 2	49 ± 8	88 ± 1

* Cytotoxic activity of WIP was not tested. Ascorbic acid was used as reference value with an inhibition of 80% at 5.9 µg/mL. Tamoxifen as anti-proliferation on the two lines of cancer cells was used as a positive control at 37 µg/mL.

**Table 4 plants-12-03231-t004:** The ABC method for preparation of Qi cha tea^®^.

ABC Process	Components of Qi Cha Tea^®^	Amount (in %)
Support	White tea	48.3
	Tulsi	12.5
	Elderberry	10.0
	Lemongrass	10.0
	Orange peel	6.5
Active ingredient (dry extract)	*Echinacea purpurea*	10.0
Hydroalcoholic solution with binder Drying	Organic acacia gum	2.5
	Blowing dry air	ND
Flavors	Natural aromas of bergamot and orange	0.2

“ND”: not determined.

**Table 5 plants-12-03231-t005:** Formulation of tea prepared by application of the Bio Concentrate Assets^®^ (ABC) method and their associated health claims from the “on hold” European botanicals list.

ABC Process	Tea Amount (%)	Botanical Amount (%)	Dry Extract Amount (%)	Health Claims (Claim ID)
Qi cha tea^®^	White tea (48.3)	*Ocimum sanctum* (tulsi) (12.5)	*Sambuscus nigra* (elderberry) (10) *Echinacea purpurea* (10)	Respiratory health: has a beneficial effect on various respiratory tract problems (ID2466); immune system health: helps the function of the natural defensive system (ID2366)

Amounts of organic lemon grass, orange peel, acacia gum, and flavors were 10%, 6.5%, 2.5%, and 0.2%, respectively.

## Data Availability

All data included in the main text.

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
