# Peer review of "Biological Activities and Polyphenol Content of Qi Cha Tea®, a Functional Beverage of White Tea Containing Botanicals and Dry Botanical Extracts with European Health Claims"

_plants, 2023, doi:10.3390/plants12183231_

Round 1

Reviewer 1 Report

The paper describe the application of the patented ABC (Bio Concentrate Assets) method with organic phytochemicals and natural aromas to white tea, to help develop a new innovative functional health beverages such as Qi cha tea, a beverage obtained either by water infusion and by pressurized liquid extraction using acacia gum as blinder.  The paper qualitatively and quantitatively reevaluate the major phytochemicals of the Qi cha tea by HPLC-DAD, and the results indicated more EGCG than the original white tea, furthermore, the authors evaluated the basic antioxidant test using the common DPPH method and cytotoxic effects against two cancer cell lines with MTT assay. This is in general a good and well written paper which therefore deserve publication, however some minor points should be addressed,

the title should be changed, too large and the contents sought in Qi cha tea are not only catechins...

- rephrase in the abstract the lines 21-22, Caco-2 is also a human cell line ;

- The whole process of HPLC-DAD method validation should be included into the paper before quantification,

- line 68 remove the hyphen

- correct cell number in line 152

- line 177, leave space

Author Response

Reviewer 1

The paper describe the application of the patented ABC (Bio Concentrate Assets) method with organic phytochemicals and natural aromas to white tea, to help develop a new innovative functional health beverages such as Qi cha tea, a beverage obtained either by water infusion and by pressurized liquid extraction using acacia gum as blinder.  The paper qualitatively and quantitatively reevaluate the major phytochemicals of the Qi cha tea by HPLC-DAD, and the results indicated more EGCG than the original white tea, furthermore, the authors evaluated the basic antioxidant test using the common DPPH method and cytotoxic effects against two cancer cell lines with MTT assay. This is in general a good and well written paper which therefore deserve publication, however some minor points should be addressed.

- the title should be changed, too large and the contents sought in Qi cha tea are not only catechins...

REPLY - the new title has been shortened and modified according to the reviewer as follows:

Biological activities and polyphenol content of Qi cha teaâ, a functional beverage of white tea containing botanicals and dry botanical extracts with European health claims

- rephrase in the abstract the lines 21-22, Caco-2 is also a human cell line ;

REPLY - this has been done i.e.,  « … in human Caco-2 and HCT-116 cell lines »

- The whole process of HPLC-DAD method validation should be included into the paper before quantification,

REPLY –  the PHLC-DAD-this has been added I.E. « The device used was a Chromatographic Fingerprint Analyses by High-Performance Liquid Chromatography Coupled with Diode Array Detector (HPLC-DAD) from Thermo Fisher Scientific, USA) as previously reported by Dawra et al. [13]. The HPLC analysis was carried out using an ultimate 3000 pump-Dionex and Thermos Separation product detector DAD model (Thermo Fisher Scientific, USA). The separation was carried out on an RPC18 reversed-phase column (Phenomenex, Le Pecq, France). The samples were filtered through a Millex syringe filter (Sigma Aldrich, Saint-Quentin-Fallavier, France). The identification of the compounds by HPLC-DAD was based on the comparison of the retention times and the DAD spectra at the maximum absorbance of each compound to those obtained with the co-injection of commercial polyphenols standarts. »

A most recent references was a new one published in Plants in 2023. «  [13] Dawra, M.; Nehme, N.; El Beyrouthy, M.; Abi Rizk, A.; Taillandier, P.; Bouajila, J.; El Rayess, Y. Comparative Study of Phytochemistry, Antioxidant and Biological Activities of Berberis libanotica Fruit and Leaf Extracts. Plants 202312, 2001. https://doi.org/10.3390/plants12102001. »

- line 68 remove the hyphen

REPLY – this has been done

- correct cell number in line 152

REPLY – this has been done i.e., “…  at 13 × 103 cells/well for HCT116 and 12 × 103 cells/well”

- line 177, leave space

REPLY – this has been done

Reviewer 2 Report

The article entitled (Biological activities and catechins content of Qi cha teaâ, a patented preparation of white tea (Camellia sinensis) with botanicals and dry botanical extracts as a functional beverage according European health claims) by Maixent et al., reported the assessments of the content of tea polyphenols and caffeine, and biochemical properties of Qi cha tea® and its botanical constituents. In addition, antioxidant and cell viability activities were determined by DPPH assay and MTT assay using Caco-2 and human HCT-116 cell lines, respectively.

The article is good and can be accepted after covering the following issues.

1-   I know the components of Qi cha teaâ but the authors should add the other plants used in this preparation in the manuscript.

2-   Authors used reference drugs for DPPH and cytotoxicity, but without mentioning their results values, so please add the values of ascorbic acid and tamoxifen in the manuscript and in Table 3, then the authors should compare the obtained data related to the standards` values.

3-   Add the standard drugs` values also in the abstract.

4-   Concerning the HPLC chromatogram, please label each peak and add the corresponding name.

5-   The Qi cha teaâ contains number of other plants, so why in the HPLC chromatogram limited number of compounds appeared??

6-   English should be carefully revised, regarding the sentence’s structures. In addition, there are many grammatic and typing mistakes that should be corrected by authors. Avoid the use of pronoun ``We``.

7-   Write the full name of all abbreviations when they are firstly appeared, e.g., ABC, MTT, EGCG, …….

8-   All plants genus and species names should be italicized throughout the whole MS.

9-   ``Camellia sinensis``, mention it first full the use C. sinensis throughout the whole MS.

English should be carefully revised, regarding the sentence’s structures. In addition, there are many grammatic and typing mistakes that should be corrected by authors. Avoid the use of pronoun ``We``.

Author Response

Reviewer 2

The article entitled (Biological activities and catechins content of Qi cha teaâ, a patented preparation of white tea (Camellia sinensis) with botanicals and dry botanical extracts as a functional beverage according European health claims) by Maixent et al., reported the assessments of the content of tea polyphenols and caffeine, and biochemical properties of Qi cha tea® and its botanical constituents. In addition, antioxidant and cell viability activities were determined by DPPH assay and MTT assay using Caco-2 and human HCT-116 cell lines, respectively. 

The article is good and can be accepted after covering the following issues. 

1-   I know the components of Qi cha teaâ but the authors should add the other plants used in this preparation in the manuscript.

REPLY - The composition of other plants, i.e., Elderberry, Tulsi, Echinacea purpurea, orange peel, lemongrass and acacia gum, has been added in different sections of the manuscript.

2-   Authors used reference drugs for DPPH and cytotoxicity, but without mentioning their results values, so please add the values of ascorbic acid and tamoxifen in the manuscript and in Table 3, then the authors should compare the obtained data related to the standards` values.

      REPLY -  To addres this concern, we have added the following footnote to Table 3: Ascorbic acid was used as reference value with an inhibition of  80 % at 5.9 µg/mL. Tamoxifen as anti-proliferation on the two lines of cancer cells was used as a positive control at 37 µg/mL.

3-   Add the standard drugs` values also in the abstract.

      REPLY – this has been done

4-   Concerning the HPLC chromatogram, please label each peak and add the corresponding name.

      REPLY - A new figure 2 has been made that has this information.

5-   The Qi cha teaâ contains number of other plants, so why in the HPLC chromatogram limited number of compounds appeared??

            REPLY - We used only the two chromatograms from the original white tea used and the Qi cha tea that result of the ABC process. On the other hand, if you look at the chromatograms there are compounds which absorb much more than others (related to the absorption coefficient and the concentration) but there are many small peaks close to the base line which correspond to compounds detectable at 280 nm.

6-   English should be carefully revised, regarding the sentence’s structures. In addition, there are many grammatic and typing mistakes that should be corrected by authors. Avoid the use of pronoun ``We``.

      REPLY – This has been done throughout the manuscript. We asked an English-speaking Canadian colleague with editorial experience to complete this process. 

7-   Write the full name of all abbreviations when they are firstly appeared, e.g., ABC, MTT, EGCG, …….

      REPLY - This has been done in the abstract and throughout the manuscript.

8-   All plants genus and species names should be italicized throughout the whole MS.

REPLY – This has been done for C sinensis and Echinacea purpurea, the other names are generic  names.

9-   ``Camellia sinensis``, mention it first full then use C. sinensis throughout the whole MS.

REPLY – this has been done

Comments on the Quality of English Language

English should be carefully revised, regarding the sentence’s structures. In addition, there are many grammatic and typing mistakes that should be corrected by authors. Avoid the use of pronoun ``We``.

REPLY – This has been done throughout the manuscript. We asked an English-speaking Canadian colleague with editorial experience to complete this process. 

Reviewer 3 Report

The article “Biological activities and catechins content of Qi cha tea tented preparation of white tea (Camellia sinensis) with botanicals and dry botanical extracts as a functional beverage according to European health claims” is devoted to the study of the chemical composition, the antioxidant activity and the cell viability or cytotoxicity activities of Qi cha tea® and all botanical constituents of Qi cha tea®.

The article may be published in Plants magazine after minor changes.

I ask the authors to pay attention to the following questions.

It is not clear what the authors wanted to say with the next sentence. "Qi cha tea® showed the lowest activity in the viability of the two cell lines."

Table captions and bibliography should be in the same style.

I ask you to give the transcript of the abbreviation ABC in the resume on line 16, and not on line 29. The abbreviation EGCG also needs to be deciphered.

The Maixent self-citation rate is 28% (refs 6-12, 17). It is necessary that this percentage be no more than 10-15%.

Line 48. What do the numbers “761” mean?

Line 59. What does "Article 13 claims" mean?

The article says that the studies were carried out twice, for example, on anticancer activity. But after all, at least three repeated measurements are necessary.

For each device used, please provide the year of manufacture, country and manufacturer.

The link to table 1 is too far from the table itself. I recommend rearranging the table closer.

What plant fragments or whole plants were used? Dried, frozen or wet?

Why exactly these plants were chosen for the composition of tea.

Water of what degree of purification was used for work.

On line 75, the abbreviation water infusion (WI) is written. On line 112, the abbreviation for water infusion products (WIP) is written. On line 177, the abbreviation WI products (WIP) is written. A similar situation is observed with PLE and PLEP. Please use the same terms. Please take a closer look at Table 3.

Line 259. I ask you to tell me about the processes using encapsulation and/or nanoparticles for tea catechin preservation. I ask you to confirm this story with literary references.

Why was Cytotoxic activity not determined for aqueous extracts.

I ask for an explanation of the value 6±6 - Antioxidant activity Cytotoxic activity* (DPPH assay) for Echinacea WI. IC50 µg/mL data are not available for all plants.

Some sentences are difficult to understand. Probably some words are missing.

Author Response

Reviewer 3

The article “Biological activities and catechins content of Qi cha tea tented preparation of white tea (Camellia sinensis) with botanicals and dry botanical extracts as a functional beverage according to European health claims” is devoted to the study of the chemical composition, the antioxidant activity and the cell viability or cytotoxicity activities of Qi cha tea® and all botanical constituents of Qi cha tea®.

The article may be published in Plants magazine after minor changes.

I ask the authors to pay attention to the following questions.

It is not clear what the authors wanted to say with the next sentence. "Qi cha tea® showed the lowest activity in the viability of the two cell lines."

REPLY – This has been completed to – « Qi cha tea® showed the lowest cytotoxic activity in the viability of the two cell lineswhen compared to white tea »

Table captions and bibliography should be in the same style.

I ask you to give the transcript of the abbreviation ABC in the resume on line 16, and not on line 29. The abbreviation EGCG also needs to be deciphered.

REPLY- This has been done.

The Maixent self-citation rate is 28% (refs 6-12, 17). It is necessary that this percentage be no more than 10-15%.

REPLY -  There are now only 3 references cited instead of 6.

Line 48. What do the numbers “761” mean?

REPLY - This is the code number  761 for the Gingko biloba extract which has a registration as drug in Europe.

Line 59. What does "Article 13 claims" mean?

REPLY - “General function” claims under Article 13.1 of the EC Regulation on nutrition and health claims refer to the role of a nutrient  or substance in growth, development and body functions; psychological and behavioural functions; slimming and weight control, satiety or reduction of available energy from the diet.

The article says that the studies were carried out twice, for example, on anticancer activity. But after all, at least three repeated measurements are necessary.

REPLY – All the analyzes were carried out in triplicate. For the anticancer activity two cell lines were tested but the tests are carried out three times.

For each device used, please provide the year of manufacture, country and manufacturer.

REPLY – the required information was added : The device used was a Chromatographic Fingerprint Analyses by High-Performance Liquid Chromatography Coupled with Diode Array Detector (HPLC-DAD) from Thermo Fisher Scientific, USA) as previously reported by Dawra et al. [13]. The HPLC analysis was carried out using an ultimate 3000 pump-Dionex and Thermos Separation product detector DAD model (Thermo Fisher Scientific, USA). The separation was carried out on an RPC18 reversed-phase column (Phenomenex, Le Pecq, France). The samples were filtered through a Millex syringe filter (Sigma Aldrich, Saint-Quentin-Fallavier, France). The identification of the compounds by HPLC-DAD was based on the comparison of the retention times and the DAD spectra at the maximum absorbance of each compound to those obtained with the co-injection of commercial polyphenols standarts.

[13] Dawra, M.; Nehme, N.; El Beyrouthy, M.; Abi Rizk, A.; Taillandier, P.; Bouajila, J.; El Rayess, Y. Comparative Study of Phytochemistry, Antioxidant and Biological Activities of Berberis libanotica Fruit and Leaf Extracts. Plants 202312, 2001. https://doi.org/10.3390/plants12102001.

The link to table 1 is too far from the table itself. I recommend rearranging the table closer.

REPLY – this has been done

What plant fragments or whole plants were used? Dried, frozen or wet?

REPLY - Dried plant fragments of leaves or fruit plants were used. The plants used were defined by the document (reference 11 in the MS) from the European commision following expert opinions from the EFSA. This has been indicated in the Table 5.

Why exactly these plants were chosen for the composition of tea.

REPLY - These plants were chosen first according the traditional used of the leaf infusion and its equivalent quantity in the leaf extract defined by the European commission evaluation (reference 11 in the manuscript) Nutrition and Health Claims Made on Foods with Regard to Nutrient Profiles and Health Claims Made on Plants and Their Preparations and of the General Regulatory Framework for Their Use in Foods.

Water of what degree of purification was used for work.

REPLY – This is an interesting comment that should be precised and added in the Materials and Methods section with the following insertion. « Drinking water was used for the ABC process done in France. In France, drinking water is water that can be consumed without risk to health. This definition of drinking water is governed by strict standards. They set the limits that drinking water must not exceed. These limits concern substances considered harmful to health in certain doses. The quality criteria for water intended for human consumption are laid down in the French Public Health Code (CSP), which covers "all water used in food businesses for the manufacture, processing, preservation or marketing of products.»

On line 75, the abbreviation water infusion (WI) is written. On line 112, the abbreviation for water infusion products (WIP) is written. On line 177, the abbreviation WI products (WIP) is written. A similar situation is observed with PLE and PLEP. Please use the same terms. Please take a closer look at Table 3.

REPLY - All abbreviations for WI or PE have been corrected with WIP and PLEP respectively

Line 259. I ask you to tell me about the processes using encapsulation and/or nanoparticles for tea catechin preservation. I ask you to confirm this story with literary references.

REPLY – this has been done with the following insertion:

Encapsulation for tea has been used to improve the flavour of the tea beverages and the bioavailability of bioactive polyphenols (15) and for nanoparticles and catechin preservation (16-17).

These literary references have been added-

15- Shi M, Shi YL, Li XM, Yang R, Cai ZY, Li QS, Ma SC, Ye, JH, Lu JL, Liang YR, Zheng XQ. Food-Grade Encapsulation Systems for (−)-Epigallocatechin Gallate. Molecules 2018; 23:445-462.

16- Peres I, Rocha S, Gomes J, Morais S, Pereira MC, Coelho M. Preservation of catechin antioxidant properties loaded in carbohydrate nanoparticles. Carbohyd Polym 2011; 86:147–153.

17- Rocha S, Generalov R, Pereira MD, Peres I, Juzenas P, Coelho MAN. Epigallocatechin gallate loaded polysaccharide nanoparticles for prostate cancer chemoprevention. Nanomedicine 2011; 6:79–87. 

Why was Cytotoxic activity not determined for aqueous extracts.

REPLY - Water infusion for tea brewing is the usual beverage in Europe. It seems to us that it is pertinent to have these data with the Qicha tea.

I ask for an explanation of the value 6±6 - Antioxidant activity Cytotoxic activity* (DPPH assay) for Echinacea WI. IC50 µg/mL data are not available for all plants

REPLY - Explanation for 6±6 : Means ± SD and IC50 is estimated if more than 50 % of inhibition is obtained at the concentration used. If less inhibition is found, the IC50 was not calculated.

Reviewer 4 Report

This study aims to determine the content of tea polyphenols and caffeine, and antioxidant and cell viability activities of Qi cha tea® and its botanical constituents. The research is interesting and the results maybe help to exploit a new innovative functional health beverage, which comply with European health claims.

However, major revision is recommended, and some other questions and suggestions are as following.

1 in section 2, only 2.1 chemicals were mentioned, samples and other materials should be provided separately in this section.

2 for the appraise of antioxidant activity, only DPPH assay was applied, it’s not enough. Generally, at least three assays should be used to evaluate in vitro antioxidant activity, please refer to the recently released publication https://doi.org/10.1016/j.lwt.2021.112740 . And in line 142, ascorbic acid was described as reference, what’s this mean? I did not see the result about this control.

3 what’s the meaning to provide chromatograms in Figure 2? Little information could be obtained from this figure. I think you could label the peaks in the chromatograms.

4 please use the standard tables in your manuscript.          

5 please delete the general or unimportant words in the key words.

6 the conclusion is too simple, and cannot summarize your research results.

Author Response

Reviewer 4

This study aims to determine the content of tea polyphenols and caffeine, and antioxidant and cell viability activities of Qi cha tea® and its botanical constituents. The research is interesting and the results maybe help to exploit a new innovative functional health beverage, which comply with European health claims.

However, major revision is recommended, and some other questions and suggestions are as following.                                           

1 in section 2, only 2.1 chemicals were mentioned, samples and other materials should be provided separately in this section.

REPLY – the following insertion was added I.E. « The device used was a Chromatographic Fingerprint Analyses by High-Performance Liquid Chromatography Coupled with Diode Array Detector (HPLC-DAD) from Thermo Fisher Scientific, USA) as previously reported by Dawra et al. [13]. The HPLC analysis was carried out using an ultimate 3000 pump-Dionex and Thermos Separation product detector DAD model (Thermo Fisher Scientific, USA). The separation was carried out on an RPC18 reversed-phase column (Phenomenex, Le Pecq, France). The samples were filtered through a Millex syringe filter (Sigma Aldrich, Saint-Quentin-Fallavier, France). The identification of the compounds by HPLC-DAD was based on the comparison of the retention times and the DAD spectra at the maximum absorbance of each compound to those obtained with the co-injection of commercial polyphenols standarts. »

A most recent references was a new one published in Plants in 2023. «  [13] Dawra, M.; Nehme, N.; El Beyrouthy, M.; Abi Rizk, A.; Taillandier, P.; Bouajila, J.; El Rayess, Y. Comparative Study of Phytochemistry, Antioxidant and Biological Activities of Berberis libanotica Fruit and Leaf Extracts. Plants 202312, 2001. https://doi.org/10.3390/plants12102001. »

« Drinking water was used for the ABC process done in France. In France, drinking water is water that can be consumed without risk to health. This definition of drinking water is governed by strict standards. They set the limits that drinking water must not exceed. These limits concern substances considered harmful to health in certain doses. The quality criteria for water intended for human consumption are laid down in the French Public Health Code (CSP), which covers "all water used in food businesses for the manufacture, processing, preservation or marketing of products.»

2 for the appraise of antioxidant activity, only DPPH assay was applied, it’s not enough. Generally, at least three assays should be used to evaluate in vitro antioxidant activity, please refer to the recently released publication https://doi.org/10.1016/j.lwt.2021.112740 . And in line 142, ascorbic acid was described as reference, what’s this mean? I did not see the result about this control.

    REPLY.           There are several principles of methods for antioxidants methods. DPPH. radical scavenging is an easy spectrophometric method and a useful method with regard to measuring the radical scavenging capacity of food constituents and plant extracts that is the case here (Gulcin, 2020). Furthermore using 3 antioxidant methods, scavenging DPPH ranked with the highest antioxidant activity among these 3 methods according the total phenolic contents (Zheng 2023). More specifically the DPPH radical scavenging activity was used to characterize different teas from green, white, oolong and black teas (He et al.  J food biochem 2020). This was the reason for the choice of DPPH. scavenging method.

    The three references cited above are now cited in the MS and that of Zheng et al 2022 proposed by the reviewer was discussed above and cited in the MS now.

- Gulcin Ä°. Antioxidants and antioxidant methods: an updated overview. Arch Toxicol. 2020 Mar;94(3):651-715. doi: 10.1007/s00204-020-02689-3. Epub 2020 Mar 16. PMID: 32180036.

-Zheng, B, Yuan, Y, Xiang, J, W, Jin, Johnson, Joel B., Zhenzhen, Li,  Wang C and Luo, D (2022). Green extraction of phenolic compounds from foxtail millet bran by ultrasonic-assisted deep eutectic solvent extraction: Optimization, comparison and bioactivities. LWT Food Sciences and technology 154 112740 112740. https://doi.org/10.1016/j.lwt.2021.112740

-He Y, Lin Y, Li Q, Gu Y. The contribution ratio of various characteristic tea compounds in antioxidant capacity by DPPH assay. J Food Biochem. 2020;00:e13270. https://doi.org/10.1111/jfbc.13270

3 what’s the meaning to provide chromatograms in Figure 2? Little information could be obtained from this figure. I think you could label the peaks in the chromatograms.

REPLY – a new version of Figure 2 is now in the MS

4 please use the standard tables in your manuscript.     

REPLY – this has been done     

5 please delete the general or unimportant words in the key words.

REPLY - Medicinal plants, tea and food have been deleted in the keywords.

6 the conclusion is too simple, and cannot summarize your research results.

REPLY – A more comprehensive conclusion has been completed.

            The ABC method (Bio Concentrate Assets®) has many advantages and offers a real breakthrough in the production of teas and botanical infusions for health and wellness. Qi cha teaâ is the white tea-based formulation produced by this innovative method. Qi cha tea® maintains its antioxidant activity and showed the lowest cytotoxic activity in the viability of the two cell lines. In conclusion, Qi cha tea® (a mixture of white tea and Elderberry, Tulsi, Echinacea purpurea, orange peel, lemongrass and acacia gum) has allowed the development a new innovative functional health beverage which complies with European health claims.

Round 2

Reviewer 2 Report

No Comments.

Minor editing of English language required

Author Response

This has been done by Pr Susan Whiting from Canada.

Reviewer 4 Report

The authors have responded to most of the comments, and revised the manuscript according to the suggestions. However, two questions are not well addressed.  

 1. the materials, including tea sample, tea polyphenol standards, should be moved to section 2.1, for example, line 117-119

 2 for the second question, the last part was not responded.

Author Response

The authors have responded to most of the comments, and revised the manuscript according to the suggestions. However, two questions are not well addressed.  

  1. the materials, including tea sample, tea polyphenol standards, should be moved to section 2.1, for example, line 117-119

REPLY-  the paragraph was moved to section 2.1  line 98. « Organic white tea, dried parts (Elderflower, Orange peel, Tulsi, Echinaceapurpurea dry extract, and Lemongrass), and natural aromas of Bergamot and Orange were obtained from Thé de la Pagode, Paris, France.”

  1. for the second question, the last part was not responded.

“The last part of  the second section was “And in line 142, ascorbic acid was described as reference, what’s this mean? I did not see the result about this control.”

      REPLY - To address this concern, this has been done with the insertion in the following footnote to Table 3:  “Ascorbic acid was used as reference value with an inhibition of  80 % at 5.9 µg/mL. Tamoxifen as anti-proliferation on the two lines of cancer cells was used as a positive control at 37 µg/mL. “

      And line 164, the concentration used of this negative control was done in the M&M section “Ascorbic acid at 5.9 μg/mL was used as reference.”
